# Phytochemical, Antioxidant, Anti-Microbial, and Pharmaceutical Properties of Sumac (*Rhus coriaria* L.) and Its Genetic Diversity

**Anna Perrone** [1], **Sanaz Yousefi** [2], **Boris Basile** [3], **Giandomenico Corrado** [3], **Antonio Giovino** [4], **Seyed Alireza Salami** [5], **Alessio Papini** [6] **and Federico Martinelli** [1,*]

1. Department of Biological, Chemical and Pharmaceutical Sciences and Technologies (STEBICEF), University of Palermo, Viale delle Scienze, 90128 Palermo, Italy
2. Department of Horticultural Science, Faculty of Agriculture, Bu-Ali Sina University, Hamedan 65167-38695, Iran
3. Department of Agricultural Sciences, University of Naples Federico II, 80055 Portici, Italy
4. Council for Agricultural Research and Economics (CREA), Research Center for Plant Protection and Certification (CREA-DC), SS 113 Km 245,5, 90011 Bagheria, Italy
5. Department of Horticultural Sciences, Faculty of Agricultural Science and Engineering, University of Tehran, Karaj 31587-77871, Iran
6. Department of Biology, University of Florence, Via Madonna del Piano 6, Sesto Fiorentino, 50019 Florence, Italy
* Correspondence: federico.martinelli@unifi.it; Tel.: +39-331-803-9998

**Abstract:** *Rhus coriaria* L., commonly known as sumac, is a shrub of the Anacardiaceae family present in various subtropical and temperate regions of the world. Considering the rich array of functional and nutraceutical ingredients, sumac extracts are an underutilized source of health-promoting dietary ingredients. For example, sumac is a spice with remarkable antioxidant activity thanks to the high presence of phenolic compounds. In addition, sumac extracts also possess antimicrobial activity and exhibit antidiabetic and hypoglycemic properties. Based on the scientific records retrieved in reliable citation databases (Scopus and Web of Science), this review comprehensively offers research results on sumac with a focus on the phytochemical profiles, the antimicrobial and antioxidant properties of the extracts, the pharmaceutical uses, and the genetic diversity. We discuss that the use of sumac as a climate-resilient tree should be promoted to diversify the food basket by leveraging on its multiple health benefits and also to reverse the abandonment of marginal lands under low irrigation.

**Keywords:** Anacardiaceae; antioxidant; pharmacology; phytochemistry; sumac; plant genetic cresources; diversity

## 1. Introduction

Nowadays, consumers have a raising confidence that food plays a central role in their well-being. Therefore, they pay more attention to high-quality food and start to be influenced not only by the appearance, taste, and flavor but also by claims related to health benefits. Although different definitions are present in the literature, from a dietary perspective, a functional food can be defined as the one able to affect beneficially one or more target functions in the human body, beyond adequate nutritional effects, in a way that is relevant to either an improved state of health and well-being and/or a reduction in risk of disease [1]. Consumers are also likely to appreciate food rich in nutraceuticals, especially if claims are supported by scientific evidence [2]. The term "nutraceutical" is the blend of "nutrition" and "pharmaceutical", and it often refers to a food component or the whole food that provides medical or health benefits, including lowering disease risk.

Plants are the most important sources of antioxidants in our diet, and the list of colorful fruits, flowers, leaves, roots, rhizomes, seeds, and bark that can provide nutritional

and functional benefits to humans is virtually endless. Wild berries of *Rhus coriaria* L. (Anacardiaceae), *Crataegus monogyna* Jacq. (Rosaceae), *Sambucus nigra* L. (Adoxaceae), *Myrtus communis* L. (Myrtaceae), and *Rubus ulmifolius* Schott (Rosaceae) are some of the most common Neglected and Underutilized Species (NUS) in the Mediterranean area. These plants, like many other NUS, have a long history of use as food, spice, traditional medicine. In some cases, they are also employed to produce specific goods (e.g., dyes, cosmetics). The growing interest in sustainable practices and vegetable foods rich in nutraceuticals determines a large attention towards the exploitation of NUS, especially in an urban-style diet [3]. The reintroduction of many of these wild herbs and berries into the diet could bring benefits not only to rural ecosystems but also to human health, thanks to their high content of minerals, vitamins, and polyphenols [4]. Recognizing the full potential of NUS by properly documenting and presenting their features, especially those linked to the nutrition and dietary value, is therefore essential to promote their conservation and use. To these aims, it will be necessary to create multi-disciplinary research networks that should span from agronomic investigation to the biochemical characterization, from the analysis of the diversity to breeding.

*R. coriaria* is an example of a nutritional fruit that could be better investigated, characterized, for instance, starting from its benefits to counteract risks related to oxidative stress and inflammation [5]. This review aims to provide valuable information on the uses, phytochemical and bioactivity properties, pharmaceutical, nutraceutical, and genetic aspects of sumac starting from a bibliographic review of the scientific literature. Our ultimate goal is to provide a comprehensive reference for further research aiming at valorizing the biological properties of this species and guiding its genetic selection.

## 2. Bibliometric Analysis and Scientific Trends

We carried out our bibliometric search in the two most relevant citation databases, Scopus (https://www.scopus.com; accessed on 21 October 2022) and Web of Science (https://www.webofknowledge.com; accessed on 21 October 2022). We used as query text "sumac" and "*Rhus coriaria*" without time limitations. We retrieved 200 and 174 documents, respectively. The total filtered and unique documents were 183 namely, 170 articles, 10 reviews, 2 book chapters, and 1 note. In addition to the removal of the overlapping documents of the two databases, the filtering parameters were the exclusion of conference papers and of non-English material. Main descriptors of our search are reported in Table 1. The bibliometric analysis indicated that the number of documents is limited (approximately, 6.3 documents per year), considering that the first publication was in 1993 and that our keyword search is expected to retrieve the whole scientific research on the selected plant species. Nonetheless, the bibliography has an interesting growth rate and an appreciable level of international cooperation (evaluated considering authorship).

**Table 1.** Main descriptor of the retrieved documents employed for this review.

| Descriptor | Result |
|---|---|
| Documents | 183 |
| Year of first document | 1993 |
| Annual growth rate | 8.26% |
| Average citation per document | 18.83 |
| Authors | 691 |
| Average authors per document | 4.34 |
| Single authored documents | 7 |
| International co-authorship | 26.23% |

The time analysis of the literature indicated that in the 1990s, the scientific output was negligible, while it steadily increased in the last ten years (Figure 1). However, the number of citations per year did not show an appreciable growth, although it is evident that strong fluctuations may reflect the relatively small number of citations per year.

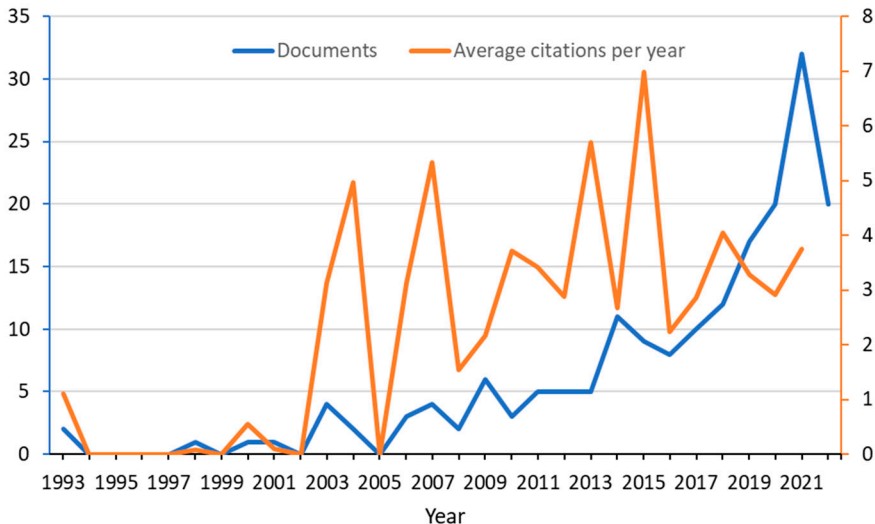

**Figure 1.** Evolution of the main bibliographic indices, number of documents (blue), and average number of citations per year (orange).

The analysis of the most relevant source revealed interesting insights (Figure 2). Most of the articles were published in journals that deal with the chemical aspects of the plant, which is consistent with the traditional use of the plant (food seasoning, dye, and in traditional medicine). Studies related to agronomic and other technical aspects to improve yield and quality traits are limited, indicating the little attention that the research community has devoted to the management and improvement of this plant species.

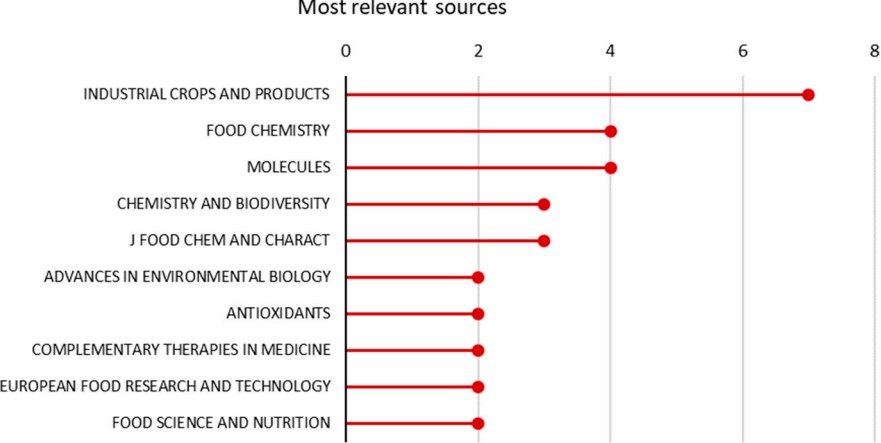

**Figure 2.** Most relevant journal sources for sumac in terms of number of publication (1993–2002). The total number of unique sources is 148.

To simplify the representation of the most used terms present in the selected scientific literature, we built a wordcloud based on the most frequently appearing "keyword plus" (Figure 3). The identified subject domains indicated the relevance of this plant species for controlled studies based on plant extract and fruits, especially to assess the potential benefit for humans. Moreover, relevant subject domains are also related to studies dealing with the chemical and biochemical (mainly antioxidant) properties.

Finally, we analyzed the relation between the scientific production and countries, based on the authors' appearances by country affiliations (Figure 4). In total, 35 countries were represented. There were large differences among the top contributing countries. For instance, there were 161 authors (out of a total of available 473) from Iran, 95 from Turkey, and 52 from Italy, but the following six countries (Lebanon, Egypt, Iraq, Jordan, Slovakia,

and Saudi Arabia) had between 20 and 11 contributions. Overall, the analysis indicated that the scientific production matches the diffusion and use of the plants (e.g., use in the Middle Eastern cuisine and to brew beverages), although significant differences are present among nations.

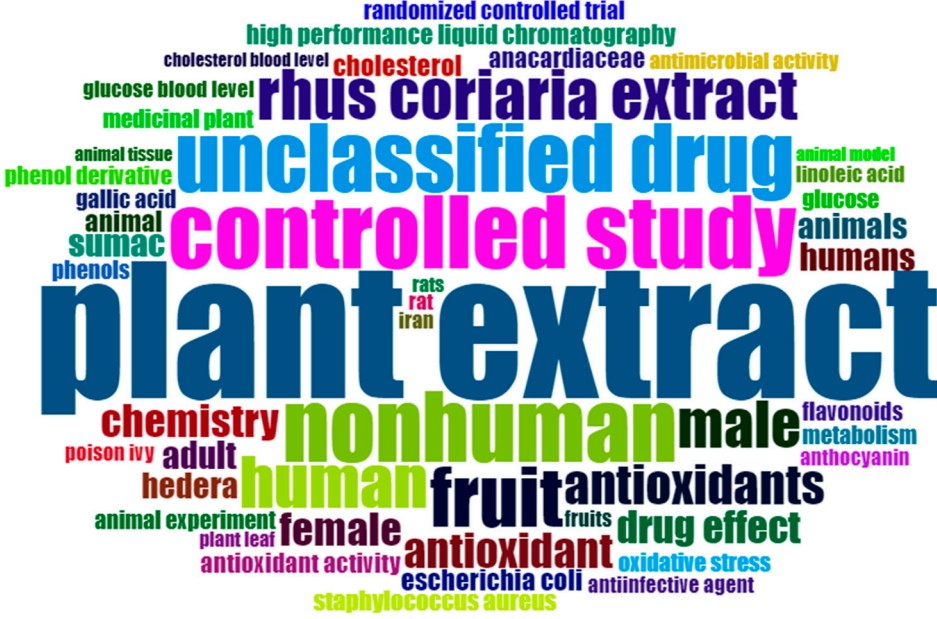

**Figure 3.** Wordcloud generated from the "keyword plus" (n = 1836) of the 183 selected documents according to the frequency of word occurrence. Keywords such as "sumac", "rhus", and "article" were omitted as too generic. "Plant extract" and "plant extracts" were considered synonymous. The illustration was created with the wordcloud2 package in R.

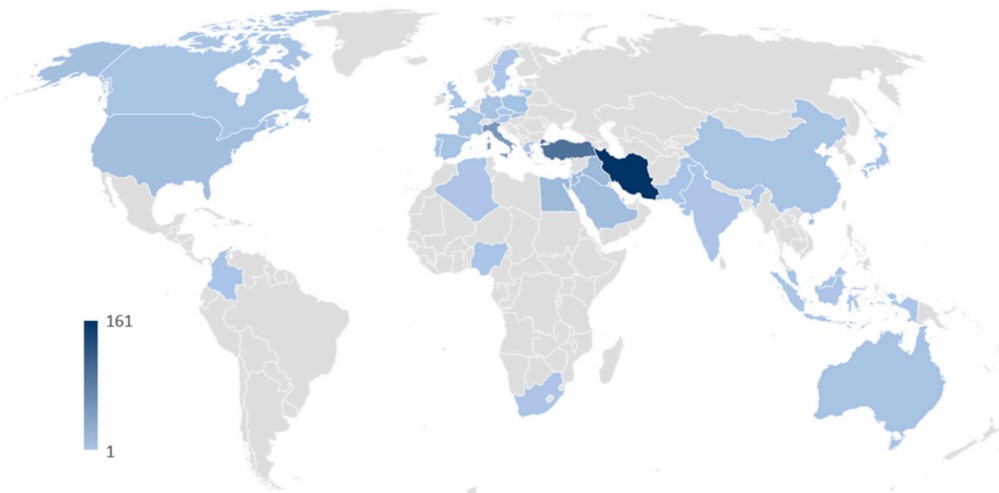

**Figure 4.** Geographical contribution to the scientific sector in terms of nationality of the authors. The color bar on the left corner illustrates the relation between the intensity of the blue color and the contributions.

## 3. Botanical Description

*R. coriaria* L. (≡ *Rhus amoena* Salisb.; ≡ *Toxicodendron coriaria* (L.) Kuntze), commonly known as sumac, Sicilian sumac, sumach, tanner's sumach, or elm-leaved sumach, is a shrub of the Anacardiaceae family belonging to the order of Sapindales. The genus *Rhus* comprises more than 250 species, typical of subtropical and temperate regions of the world, such as Africa, the countries around the Mediterranean basin, Western Asia, and Iran. The

sumac is a shrub or small tree distributed in the natural habitats of the Mediterranean regions [6] and can grow in poor and dry soils [7] (Figure 5). The plant is characterized by spirally arranged deciduous leaves and small flowers present in a spike/dense panicle (5 to 30 cm in length) [8]. The flowers have five petals from greenish-white to creamy-white or red petals [9]. Sumac fruits are oval/round, hairy drupes, with a mean weight of 0.014 g/fruit, and a mean length and width of around 3.88 mm [10]. A fruit contains one brownish seed, usually of 0.2–0.5 cm in diameter and length. Fruits are formed in dense clusters (100–700 fruits/inflorescence) and are normally of a deep red color [11]. Sumac fruits contain citric and malic acid, which provide a sour taste, and are dried to be used as spice [12]. Sumac can be propagated through seeds and rhizomes, and artificial propagation is carried out mainly using sprouts from rhizomes.

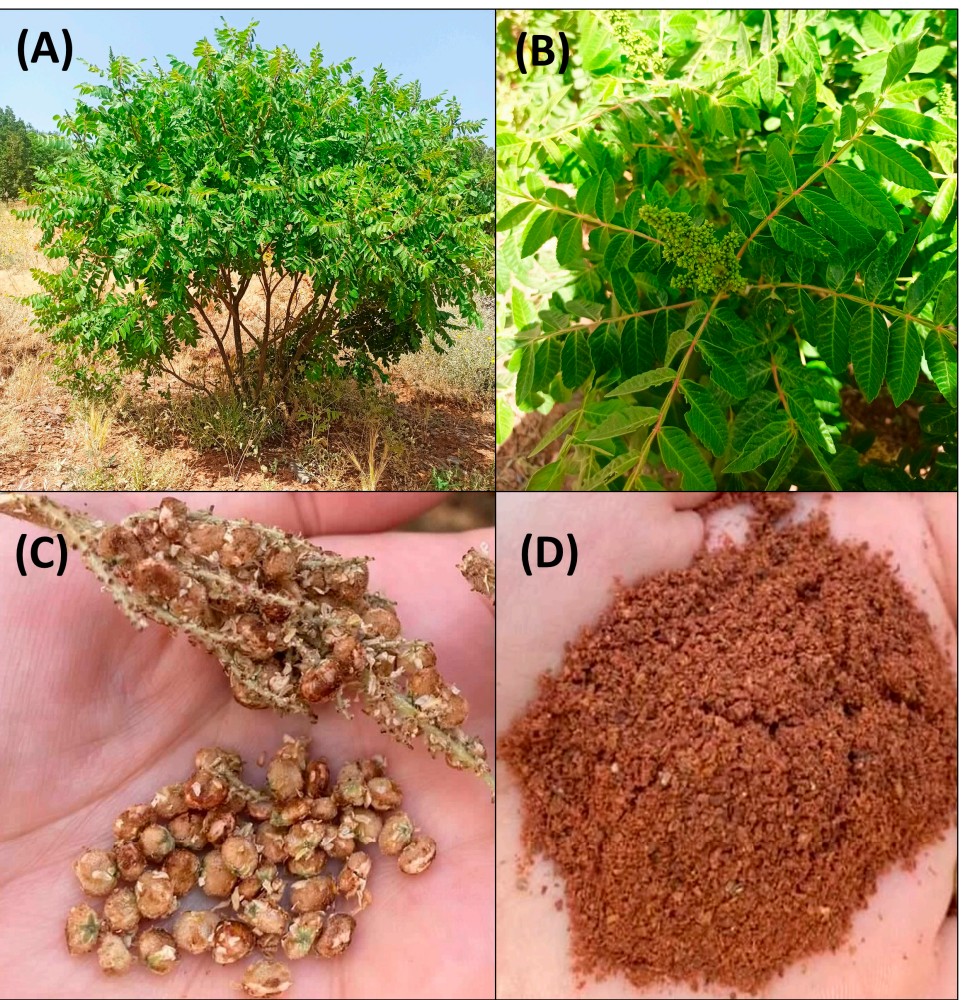

**Figure 5.** *Rhus coriaria* L. (sumac) morphology: (**A**) shrub; (**B**) leaves and immature inflorescence; (**C**) dried fruits; (**D**) powdered fruits (spice).

## 4. Uses and Phytochemistry

The medicinal value of sumac was first described about 2000 years ago by the Greek physician and botanist Pedanius Dioscorides (40–90 AD) [13]. *R. coriaria* has been employed as food and feed [14–17]. Sumac is also used in the textile and leather industry because of the dyeing properties of the bark [18]. The fruits are usually dried and ground to yield a dark red powder, used as a spice to provide a pleasant acid and astringent taste. The spice is used in the Mediterranean and in the Middle East countries such as Syria, Jordan, Turkey, and Iran [19]. Gastronomically, sumac is used as a flavoring additive in a large variety of recipes (meats, fish, chicken, egg, and salads) but also to brew beverages.

Another use of sumac is as a traditional medicine, for instance, to treat conditions such as liver diseases [20–22], urinary system issues [20], ulcers [23]. Recently, silico studies predicted that the sumac phytochemicals can inhibit COVID-19 [24]. Pulverized fruits have also been used to increase sweating and lower cholesterol levels [22]. These therapeutic uses are generally attributable to its various biological properties such as antioxidant, anti-inflammatory, and hypolipidemic hypoglycemic activities [25].

Sumac is potentially a high nutritional source of useful compounds, due to the richness of phenolics, tannins, flavonoids, and organic acids. Major phenolic compounds are cyanidin glucoside, delphinidin glucoside, and delphinidin [26]. Among the flavonoids and malic acid derivatives, flavonols such as myricetin, kaempferol, and quercetin are usually the most abundant [27]. Sumac fruits also contain thiamine, riboflavin, cyanocobalamin, pyridoxine, biotin, nicotinamide, and ascorbic acid [7] (Figure 6).

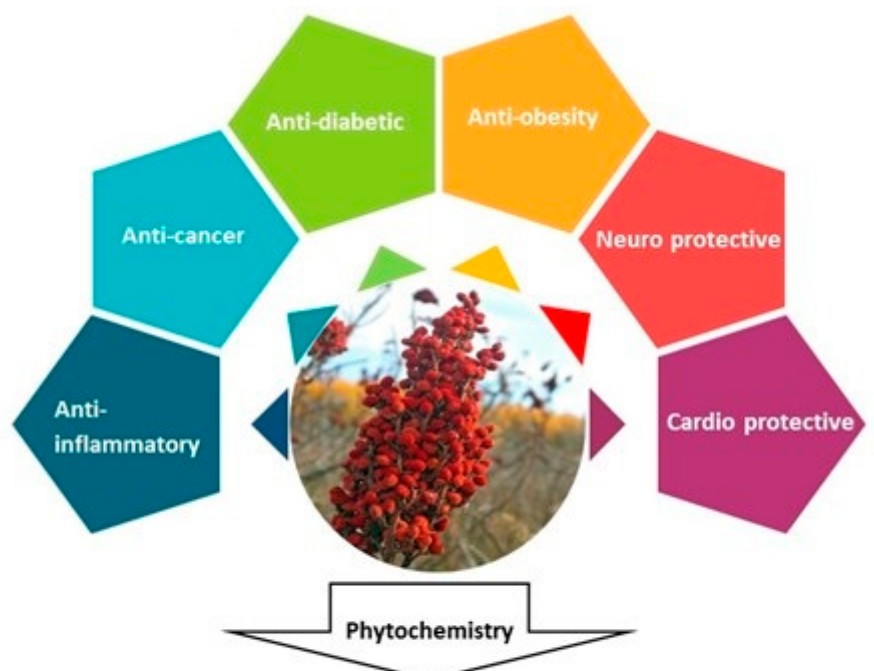

**Figure 6.** A graphical illustration of the health-promoting activities attributed to the *Rhus coriaria* fruit. The artwork also reports main bioactive compounds.

Among the secondary metabolites, aqueous, alcoholic, and lipid extracts from leaves and fruits are characterized by a high percentage of hydrolysable tannins, followed by flavonoids [28]. Anthocyanins such as coumarate moieties phenolic acids (gallic, protocatechuic, p hydroxybenzoic, and vanillic acids), phenolic acid methyl esters, and galloyl glucoses are present in the fruits [29]. Hydrolyzable tannins represent approximately 20% of the fruit mass. Phenolic compounds such as gallic acid and quercetin were identified in sumac using HPLC. Galactotannins which are used in foods as stabilizers are also abundant [30]. Gallic acid, one of the major constituents in sumac fruit, has been shown to reduce $H_2O_2$-induced DNA damage in human lymphocytes [31]. The extraction solvent is crucial to isolate polyphenols, and its choice has obvious industrial implications. Ethanol (80%) extraction proved to be a suitable, relatively toxic solvent for extracting polyphenolics from fruits [32]. Other studies show that methanol extraction from fruits can provide the highest total phenolic content (151.71 mg g$^{-1}$), followed by ethyl acetate (65.31 mg g$^{-1}$) and aqueous extracts (6.10 mg g$^{-1}$) [33–35]. In leaves, maceration in water at 45 °C for 60 min

resulted in an improved extraction rate of phenolic compounds [11]. The phytochemical analysis of the ethanolic extract of the seeds indicated the abundance of phenolic compounds and the presence of compounds with antifungals (*Candida albicans* and *Aspergillus flavus*) [36]. Finally, the analysis of the essential oils (EOs) from different organs (e.g., stem, buds, leaves, flowers, and fruits at different stages) indicated that monoterpenes and sesquiterpenes were the most abundant chemical classes. In particular, β-caryophyllene and α-pinene were the predominant compounds except for the EO of the leaves which were cembrene and β-caryophyllene and were most abundant [37].

## 5. Antioxidant and Protective Activity

Sumac can gain the reputation of a useful food plant because of the antioxidant activity linked to the presence in fruit of phenolic compounds (Table 2). The antioxidant capacity of the methanol extract of the dried fruits was used to counteract the autoxidation and lipolysis of oil samples compared with a reference commercial synthetic antioxidant compound, such as butyl hydroxyanisole (BHA) [38]. The sumac extracts limited the formation of hydroperoxide, such as BHA, but the antioxidant effects of the extracts decreased significantly after four weeks of storage. Sumac extract has also a stronger antioxidant activity than butylated hydroxytoluene (BHT), implying that it could be added as a complementary natural antioxidant and preservative to food [39].

Studies on the effect of sumac as an extract, juice, and tablets in animals and humans started to appear in the literature since 2012. Capcarova et al. [40] showed that sumac use raised the total antioxidant status (TAC) and cholesterol levels in adult male rabbits. In research conducted on type 2 diabetes patients, an increase in TAC and a reduction in serum glucose, glycosylated hemoglobin (HbA1c), apolipoprotein-B (apo-B), and apolipoprotein-AI (apo-AI) was obtained at the end of the dietary intervention with sumac [41]. A dietary study conducted on healthy volunteers used sumac juice in order to evaluate its beneficial effects on the performance of athletes. Parameters such as creatine kinase (CK), lactic acid dehydrogenase (LDH), troponin I, and hydroxyproline (hyp) were improved 30 days after drinking a sumac juice, and overall, the fruit juice was able to relieve the muscle pain after exercise [42].

Obesity-induced oxidative stress is associated with the development of various pathological conditions [43]. The anti-obesity and antioxidant activity of *R. coriaria* was first investigated by Jamous et al. proving that *R. coriaria* leaves and fruit epicarp exhibited potent inhibition of pancreatic lipase (PL) activity in vitro [44]. At the lowest doses, the antioxidant activity of *R. coriaria* was higher than *Punica granatum*, *Sarcopoterium spinosum*, and *Ceratonia siliqua* [44]. PL is the most important lipase and plays an important role in the conversion of triglycerides into monoglycerides and free fatty acids [45]. Therefore, the antioxidant and PL inhibition of sumac extract can be useful to fight obesity [5].

The methanolic extract of sumac leaves exhibited a high antioxidant and scavenging activity [46]. *R. coriaria* extracts prevented or reduced the spread of skeletal muscle atrophy by reducing oxidative stress with superoxide dismutase 2 and catalase-dependent mechanisms [47]. The protective role of the ethanolic sumac extract against damage caused by ultra-violet A light (UV-A) has been also described. The extracts also reduced the formation of DNA lesions in cells exposed to medium UV-A doses [48]. The molecular mechanism by which the extract exerts its antioxidant and genoprotective effects have yet to be clarified [5,48]. A protective effect on DNA of *R. coriaria* was also studied using human lymphocytes. Fruit extract was able to prevent $H_2O_2$ and (±)-anti-benzo[a]pyrene-7,8-dihydro-diol-9,10-epoxide (BPDE)-induced DNA damage [31]. This protective effect was confirmed in an in vivo assay with rodents. Drinking water supplemented with sumac extract significantly decreases DNA damage in all rats' organs (colon, liver, and lung) of whole-body irradiated rats [31].

## 6. Antibacterial Activity

Beyond the antioxidant activity, several studies were carried out on the evaluation of the antimicrobial activity of *R. coriaria* and its extracts. The aqueous, ethanol, and methanol extracts of sumac were all associated with a detectable antimicrobial activity. The inhibitory effect of alcohol extracts of *R. coriaria* has been studied on Gram-positive and Gram-negative foodborne and pathogenic strains. The results indicated that Gram-positive bacteria are usually more sensitive, although there are significant species-specific differences [49]. From an applied perspective, aqueous extracts of sumac improved the conservation of poultry (i.e., the refrigerated shelf life of chicken wings), and this has been associated with good antimicrobial activity against coliforms [50]. The antibacterial activity of *R. coriaria* was also studied in combination with *Thymus vulgaris* L. (Lamiaceae). The different extracts from these plants (e.g., hot water, 80% methanol, 80% ethanol) typically showed an additive inhibitory effect on *Pseudomonas aeruginosa* (Schröter) determined using an agar diffusion method [51]. An 80% ethanol extract effectively prevented the growth of *B. cereus* and *Staphylococcus aureus* (Rosenbach), two of the most common microorganisms causing food poisoning [12]. Antibacterial activity has been connected with antioxidant activities [52]. For instance, a 20% ethanolic extract of sumac inhibited *B. cereus* and *Helicobacter pylori*, and this was related to the scavenging activity [52]. Aqueous extracts of sumac also exhibited antimicrobial activity against *Streptococcus sanguinis* (White and Niven), *S. sobrinus* (Coykendall), *S. salivarius* (Andrewes and Horder), and *Enterococcus faecalis* (Andrewes and Horder) [53]. The methanol extract showed an important inhibitory activity against *Streptococcus mutans*, one of the most important microorganisms involved in tooth decay [54]. This activity was associated with the influence of methyl gallate on the bacterium adherence [54]. The essential oil of *R. coriaria* also has antibacterial properties. For example, the oil extract prevents the growth of *P. aeruginosa*, *E. coli*, and *S. aureus* or *B. subtilis* [55]. Likewise, ethanolic extracts from ground and unground sumac fruits inhibit a range of Gram-positive and Gram-negative bacteria [56].

All these findings reinforce the traditional use of various formulations of *R. coriaria* as a disinfectant. They also prompt for further research towards the isolation of antimicrobial molecules that could be used to treat microbial infections or used for the development of biopharmaceuticals [5].

## 7. Pharmaceutical and Therapeutic Activity

Berries of sumac are recognized in various cultures as a natural remedy for different types of diseases or disorders [38]. They are used in folk medicine because of the astringent properties of sumac fruits, with anecdotal evidence pointing towards a possible remedy for hypertension, diarrhea, diabetes, inflammations of the oral cavity, high blood pressure, gastro-intestinal disorders, ulcer, stomachaches, hyperglycemia, and atherosclerosis too.

A range of pharmacological and biological activities have been linked to different parts of *R. coriaria*. They include antidiabetic, cardioprotective, antidyslipidemia, antinociceptive, neuroprotective effects as well as a positive impact for dental protection [5]. One of the first studies on a potential hypoglycemic effect was on extracts of dried and ground sumac fruits (methanol, n-hexane, and ethyl acetate extracts), and it was based on the assessment of the α-amylase inhibitory activity [57]. The methanol extract showed 48.3% inhibition of α-amylase, while the ethyl acetate extract inhibited the α-amylase by 87% [57]. It was proposed that tannins and especially flavonoids could be responsible for the inhibitory activity [57]. Antidiabetic properties of ethanol extract of fruits of *R. coriaria* were later demonstrated in rats. Short- (postprandial) and long-term reductions in the blood glucose level (around 25%) were recorded following treatment with ethanol extracts [58]. In addition, the sumac extracts significantly increased high-density lipoprotein (HDL) and decreased low-density lipoproteins (LDL), superoxide dismutase (SOD) activity, and catalase (CAT) activity [58]. The effects of the methanol extract of *R. coriaria* dried seeds were studied on non-insulin-dependent diabetes mellitus rats [59]. This work indicated that the high doses of the extract decreased the levels of glucose in the blood, delaying

hyperinsulinemia and glucose intolerance in a manner comparable to drugs (i.e., pioglitazone) [58]. A protective effect of the freeze-dried extract of sumac was demonstrated in relation to the diabetic complications induced by streptozotocin (STZ). A reduction in HbA1c and an increase in the components of the antioxidant defense system (ADSC) and malondialdehyde (MDA) were observed [14].

Benefits of sumac on cholesterol levels have also been described. Early in vivo studies showed a decrease in lipid and cholesterol parameters in hypercholesterolic rats after a diet with sumac methanol extract [60]. A sumac-based diet administered to adult male rabbits improved albumin, bilirubin, and total blood antioxidant status [40]. These activities can be ascribed mainly to polyphenols because they reduce the absorption of intestinal cholesterol and increase the excretion of bile acids/salts [61]. In addition, sumac contains limonene, which is hydroxylated in perillic alcohol. This compound inhibits the reducing activity of 3-hydroxy-3-methylglutaryl coenzyme A (HMG-CoA). This may lead to an inhibition of the synthesis of farnesyl pyrophosphate, an enzyme involved in the production of cholesterol and other molecules for intracellular signaling. The impact of sumac was also determined in relation to the oxidative stress due to excess fatty acids in the diet [62]. It is known that a diet too rich in fatty acids (and cholesterol) ultimately increases the risks for the onset of atherosclerosis and thrombosis. The addition of sumac to the diet was associated with a reduction in total cholesterol, LDL cholesterol (LDL-C), fibrinogen, and oxidative stress markers in rabbits, compared to the high-cholesterol diet group. However, the consumption of sumac did not significantly affect the lowering of circulating triglycerides (TG), ApoB100 (which transports cholesterol to LDL), and factor VII [62]. Total cholesterol (Total-C), LDL-C, and TG levels in the serum of humans were reduced due to the administration of sumac to young dyslipidemic adolescents for a month. Yet, likely due to the short period of sumac consumption, its effect on HDL cholesterol (HDL-C) was not important [63]. A significant increase in HDL levels was observed in 80 patients affected by hyperlipidemia [64]. The same research team, in a more recent study, highlighted important increases in HDL-C and Apo-A1 levels in response to sumac supplementation in patients with hyperlipidemia [65]. Recent research showed that the consumption of sumac can decrease cardiovascular risk factors in people with mild to moderate hyperlipidemia thereby reducing the body mass index (BMI), blood pressure, and total cholesterol. However, the triglycerides remained unchanged [66].

The positive effect of sumac on domesticated animals was demonstrated on poultry. Sumac fruit powder improved the performance of broiler chickens growing under heat stress conditions during the first 21 days but not after the whole experiments [67]. This study also indicated that the lowest dose tested (0.5% of powder) caused this positive effect, while the highest (1%) did not. This phenomenon was discussed considering the role of tannins [67]. Tannins, as other plant-derived polymeric phenolics commonly found in fruit trees [68–71], have also been associated with antinutritional activity [72], and, when present over a certain amount, they can also add negative attributes to foods (e.g., excessive astringency). Although the type, amount, degree of condensation, and presence of other molecules can strongly affect the role of tannins in food, tannins can decrease palatability, voluntary intake of food, and utilization of nutrients. For instance, the content of tannins is relevant for animal feeding because they can decrease digestibility of protein and uncooked dry matter. These aspects are expected to be less relevant when sumac is used mainly as a food spice.

Some positive or promising effects of sumac extracts on apoptosis and cancers have been also reported. Sumac extract improved the histological features (caspase-9 immunostaining of the small bowel) in a rat Necrotizing enterocolitis (NEC) model. NEC is a disease that affects premature or low-weight infants causing severe morbidity [73]. Nonetheless, relevant clinical parameters did not significantly differ between the NEC rats fed with or without the sumac extract [73].

The ethanolic extract of dried sumac fruits had a cytotoxic effect against breast cancer cells that depended on the promotion of cell growth inhibition, cell cycle arrest, cellular

senescence, apoptosis, and autophagic cell death [74]. Using chicken embryos, the same research group showed that the ethanolic extract suppressed tumor growth and metastasis in vivo. This type of extract also inhibited the viability and growth of colorectal cancer cells (i.e., HT-29 line) and slowed tumor growth in a xenografted (HT-29) mouse model [75].

An improved recovery from inflammation is another effect attributed to sumac extracts. A positive effect of *R. coriaria* fruit extracts, especially the macerated ethanol extract (mERC), was demonstrated in the treatment of keratinocyte inflammation through their inhibitory effect on the production of skin pro-inflammatory mediators [76]. Chronic inflammatory-related diseases represent nowadays the most significant cause of death worldwide [77]. The anti-inflammatory activity of *R. coriaria* was reported by several groups. The authors demonstrated that alcoholic extract of sumac fruits significantly reduced the level of mRNA on the pro-inflammatory cytokines IL-18 and IL-1β, in lipopolysaccharide-stimulated synoviocyte extracted from the joint and fluid of limb of the 8-month-old healthy calf [78]. Neurodegenerative diseases can be initiated by ischemia, severe acute traumatic injuries, hypoxia generating oxidative stress, and neuroinflammation, which is linked to one of Alzheimer's diseases such as optic neuropathies [79]. Therefore, it is necessary to block or slow the death of neuronal cells in neurodegenerative diseases. A study was conducted in the rat retinal ganglionic cell line RGC-5 for the neuroprotective effects of the ethanolic extract of sumac extract against retinal degeneration in vitro [80]. The results showed that the extracts significantly reduced cell death induced by serum deprivation of RGC-5 cells. Furthermore, the reduction in glutathione-S-transferase (GST) and glutathione (GSH) levels induced by serum deprivation from *R. coriaria* ethanol extract (ERC) was significantly reduced [80]. Sumac fruit extract accelerates wound healing induced in Wistar rats. Specifically, the treatment with sumac (5 mg mL$^{-1}$ and 10 mg mL$^{-1}$) promoted wound healing through an increased deposition of hydroxyproline and collagen [81]. It was proposed that these effects are due to the anti-inflammatory activity of the sumac extract [81]. In isolated rat hepatocytes, the extract and one of its key components (gallic acid) protected against various sources of cellular oxidative stress [82]. A dose-dependent cardiovascular protective effect in isolated rabbit hearts was exerted by the methanolic extract [83]. The level of glucose and cholesterol in type II diabetic male mice induced by nicotinamide streptozotocin is reduced by the administration of hydroalcoholic sumac seed extract. Furthermore, LDL cholesterol levels decreased while leptin levels increased significantly in those mice when treated with a dose of 300 mg kg$^{-1}$ of hydroalcoholic extract [17].

Overall, the positive effects of sumac extracts on various pathological conditions imply that this plant species can be an important source of therapeutic compounds.

**Table 2.** Biological activities and pharmacological properties of fruit extracts of sumac (*Rhus coriaria* L.).

| Pharmacological Properties | Solvent | Activity | Ref. |
|---|---|---|---|
| Antioxidant activity | Methanol extract | Strong antioxidant activity, such as BHA. Strong antioxidant activity. | [38] |
| | | Moderate inhibiting effect of lipid peroxidation compared to synthetic antioxidants. | [29] |
| | | Higher antioxidant capacity and reducing power than ascorbic acid. | [35] |
| | Sumac juice | Beneficial effect on muscle performance among athletes in oral administration of sumac juice. | [42] |
| | Ethyl acetate EtOAc fraction | Inhibition or slowing down the progress of skeletal muscle atrophy (human myoblasts) by decreasing ROS via SOD and catalase-dependent mechanisms. | [47] |
| | Ethanol extract | Acts as a cell cycle inhibitor or apoptosis inducer in endothelial cells (HMEC-1) subjected to UV-A damage and ROS onset. | [48] |

**Table 2.** *Cont.*

| Pharmacological Properties | Solvent | Activity | Ref. |
|---|---|---|---|
| | Ethanol and methanol Extracts | Antibacterial activity of Gram positive (*B. cereus, B. megaterium, B. subtilis, B. thuringiensis*). | [49] |
| | Water extract | Stronger antioxidant activity than BHT. | [39] |
| | | Antimicrobial activity against coliform. | [50] |
| | Dietary sumac | Increasing the TAC and cholesterol levels in adult male rabbits. | [40] |
| | | Increasing TAC and decreasing serum glucose, HbA1c, apo-B, apo-A1 in diabetic patients. | [41] |
| Antibacterial activity | Ethanol 80% extract | Antibacterial activity on both Gram-positive and Gram-negative bacteria. | [12] |
| | Ethanol 20% extract | Strong antimicrobial activity against *B. cereus* and *H. pylori*. | [52] |
| | Water extract | Antimicrobial activity against Streptococcus mutans, *S. sanguinis, S. sobrinus, S. salivarius,* and *E. faecalis*. | [53] |
| | Methanol extract | Strong antimicrobial activity against *Streptococcus* mutans. | [54] |
| Antidiabetic activity | Methanol extract after fractionation with ethyl acetate and hexane | Significant hypoglycemic activity through α-amylase inhibition. | [57] |
| | Lyophilized extract | Antidiabetic activity in diabetic rats induced by STZ (lower) HbA1c, ADSCs (higher), MDA (lower). | [14] |
| | Ethanol 96% extract | Higher HDL, SOD, CAT. Lower LDL, maltase, and sucrase activities. | [58] |
| | | Reduction in postprandial blood glucose (PBG) by 24% (at 5 hr, acute) in rats. Significantly lower PBG (by 26%) and higher SOD and CAT in the long term (21 died). | [58] |
| Lipid-lowering and hypocholesterolic activity | Methanol extract | Reduction in serum lipid levels in hypercholesterolemic rats. Reversing hypertrophic cardiac histology. | [60] |
| | Dietary supplement sumac fruits, and 80% methanol extract | Decrease in cholesterol in the blood of rabbits. | [40] |
| | A fat diet with 2% of sumac powder | A significant decrease in total cholesterol, LDL-C, fibrinogen, and oxidative stress markers, compared to the high-cholesterol diet group of rabbits. | [62] |
| | Dietary sumac | Reduction in serum Total-C, LDL-C, and TG levels, one month of administration of *R. coriaria* to young dyslipidemic adolescents. Administration; its effect on HDL-C was not significant. | [63] |
| | | A significant increase in serum HDL cholesterol levels in the sumac group of eighty patients with hyperlipidemia. | [64] |
| | | Decreasing the cardiovascular risk factors in persons with mild to moderate hyperlipidemia (reduced BMI, blood pressure, and total cholesterol; triglycerides remained unchanged). | [66] |
| | | Significant increases in HDL-C and Apo-A1 levels in response to sumac supplementation in patients with hyperlipidemia. | [65] |
| | | Reduction in blood TC, VLDL-c, and FBS concentrations in broiler chicken. | [61] |
| | | Beneficial effects on broilers reared under stressful conditions. | [67] |

Legend: BMI: body mass index; ROS: reactive oxygen species; TAC: total antioxidant capacity.

## 8. Genetic Characterization

*R. coriaria* has been scarcely investigated at the genetic level. To our knowledge, there is no scientific investigation on the ploidy, dimension, and structure of the genome. In the National Center for Biotechnology Information (NCBI) database, there are only 23 nucleotide sequences associated with the query "*Rhus coriaria*", of which 16 are from this species. Limited information is present for phylogenetically close species, such as *Rhus cinensis* Mill. (197 sequences), *Rhus typhina* L. (43), *Rhus copallinum* L. (31), *Rhus glabra* L. (29),

*Rhus virens* Lindh. Ex A. Gray (24), *Rhus aromatica* L. (18) (https://www.ncbi.nlm.nih.gov/, accessed on the 4 August 2022). Moreover, the chloroplast genome of *Rhus chinensis* is available [84]. Studies dealing with the genetic diversity of sumac are also limited in number and in terms of geographical representation. Thirty genotypes of sumac from three locations in Jordan were characterized using AFLP markers [85]. The molecular genetic characterization of 24 sumac genotypes of Turkey was conducted with 17 Sequence-Related Amplified Polymorphisms (SRAP) and 12 Inter Simple Sequence Repeats (ISSR) [86]. Besides the above-mentioned studies on the chemo-diversity of sumac, research was conducted also to describe the morphological diversity of the species. For example, 136 Iranian genotypes were screened using 42 morphological traits. The observed phenotypic variation was considered sufficient to select superior accessions for fruit production [10]. The analysis of ten vegetative traits was employed to describe the morphological variation across different environments, which was described as high for most of the traits [87]. Interspecific diversity in *Rhus* was analyzed by RAPDs [88]. This work revealed two major groups: *Rhus gerrardii* (Harv. Ex Engl.) Schonl., *Rhus glauca* Thunb., *Rhus pentheri* Zahlbr and *Rhus natalensis* Bernh., and *Rhus gueinzii* Sond.

To start a successful breeding program in the *Rhus* genus and/or sumac, it would be highly desirable to understand the relation between genetic and phytochemical diversity, a crucial point to select pre-breeding material and to define targets and ideotypes. Recently, 18 ISSR primer pairs were used in an association study of phytochemical traits (e.g., polyphenols) of 75 accessions from five regions of Azerbaijan [89].

Overall, investigations on the genetic features of sumac are much more limited than the phytochemical profiling, and they have been carried out exploiting anonymous DNA markers. The limited genomic and genetic information is probably the result of the emphasis given to the phytochemical and nutraceutical properties of the plant. In this scenario, it is not easy to prioritize interventions. A first necessity would be to carry out an internationally coordinated action to assess not only the level of genetic diversity and population differentiation but also the possible genetic erosion in different countries. Moreover, it would be necessary to establish curated core collections. A third need is to start to associate the level and type of DNA diversity with agronomically and biochemically relevant traits. At the genomics level, an affordable approach would be to build a reference transcriptome, a resource that could greatly facilitate the elucidation between the rich phytochemical profile and environmental factors.

## 9. Conclusions

Many NUS in the Mediterranean area have a range of interesting pharmaceutical and nutraceutical properties, but these are not always well documented. The review of the botany, biochemistry, and health-related properties of *R. coriaria* highlighted the remarkable attributes of this species among NUS. Briefly, sumac is a rich source of tannins, polyphenols, flavonoids, and organic acids with antiviral, antimicrobial, antifungal, antidiabetic, and hypolipidemic activities. There is a good body of evidence of the potential pharmaceutical and nutraceutical properties of this species, with growing scientific research able to provide an explanation on the various health-related claims that can be important to increase consumer acceptance. On the other hand, our literature survey indicated the limited agricultural research and genetic investigation (including breeding) that sumac has experienced. Deeper insights into sumac biodiversity and biochemistry may lead to a better understanding of the molecular mechanisms underlying the synthesis of metabolites involved in its functional pharmaceutical and nutraceutical properties. This may provide the scientific background for the selection and breeding of new varieties. The adaptive properties of the species and its ability to grow under different environmental conditions, along with the development of sustainable agronomic practices, will allow promoting the use of sumac in a biodiversity-based agriculture focused on the production of diverse and healthy plant food [90].

**Author Contributions:** Conceptualization, F.M., A.P. (Anna Perrone); writing—original draft preparation, A.P. (Anna Perrone), B.B., G.C., S.Y., A.P. (Alessio Papini), S.A.S., F.M.; writing—review and editing, A.G., A.P. (Anna Perrone), B.B., G.C., S.A.S. and F.M. All authors have read and agreed to the published version of the manuscript.

**Funding:** This research received no external funding.

**Data Availability Statement:** Not applicable.

**Conflicts of Interest:** The authors declare no conflict of interest.

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
