# Peer review of "Phytochemical, Antioxidant, Anti-Microbial, and Pharmaceutical Properties of Sumac (Rhus coriaria L.) and Its Genetic Diversity"

_horticulturae, doi:10.3390/horticulturae8121168_

Round 1
Reviewer 1 Report
I have the following recommendations for the article. In the introduction of the manuscript, the terms functional foods, nutraceuticals, nutraceutical food are used. Authors should explain to the reader the differences between these terms. I recommend adding synonyms for both the Latin name and the English name of this plant. Why is the term wild berries and not fruits used in the title of the article? The text contains capital letters at the beginning of words in several places where they are not used, e.g. names of substances, etc., e.g. line 182, 195, 310-311, 376, 393 Line 178 should correctly read p-hydroxybenzoic... In the description of the plant, I would recommend adding, for example, the size of the fruits or their weight and their number, for example, per inflorescence. The article mentions the positive effects of the substances, due to the high content of tannins, but I would also recommend mentioning the possible anti-nutritional effects of other substances as well. The authors refer to sumac as a superfood, what are the possibilities of using it as a food besides spices? I consider picture number 1 to be unnecessary, because the effects of skupka are better summarized in picture number 2. In picture number 2, quercetin, myricetin and kaempferol should not be listed separately, as they belong to flavonoids. I therefore recommend placing them, for example, in brackets after the term flavonoids. in the same way, the mentioned acids would then deserve the title of phenolic acid. I also recommend editing the table, deleting the titles that are still repeated in the first column and leaving once antioxidant activity, 1 x antimicrobial, 1 x Antidiabetic, etc. Then leave once methanol extract, ethanol, etc. in the second column. and in the third column then combine the effects for these extracts or form of sumac. /dumbbells should not be written in sentences, but only passwords, as is the case at the end of the table. I would recommend adding a table with the content of specific identified substances, if known, indicating the part of the shell in which the substance was described.
Author Response
Reviewer 1:
I have the following recommendations for the article. In the introduction of the manuscript, the terms functional foods, nutraceuticals, nutraceutical food are used. Authors should explain to the reader the differences between these terms. I recommend adding synonyms for both the Latin name and the English name of this plant.
Authors response:
We rephrased and simplified the whole part and added the definitions of “functional food” and “nutraceuticals” (Lines 42-51).
In addition, we added the additional English common names given to Rhus coriaria and the two scientific homotypic synonyms (Rhus amoena; Toxicodendron coriaria) (Lines 147-149).
Why is the term wild berries and not fruits used in the title of the article?
Authors response:
We thank the reviewer for the suggestion that improved the clarity of our title. We changed the title as suggested and improved its style.
The text contains capital letters at the beginning of words in several places where they are not used, e.g. names of substances, etc., e.g. line 182, 195, 310-311, 376, 393 Line 178 should correctly read p-hydroxybenzoic...
Authors response:
We corrected the typos in the manuscript.
In the description of the plant, I would recommend adding, for example, the size of the fruits or their weight and their number, for example, per inflorescence.
Authors response:
We added the requested information about size and weight of the fruits in Ls 157-161).
The article mentions the positive effects of the substances, due to the high content of tannins, but I would also recommend mentioning the possible anti-nutritional effects of other substances as well.
Authors response:
We added, as suggested, a new section dedicated to the anti-nutritional effects of tannins in relation to other substances. It was also specified that tannins are common in several edible fruits and beverages.
The authors refer to sumac as a superfood, what are the possibilities of using it as a food besides spices?
Authors response:
To avoid misunderstanding about food and spices we rephrased the whole sentence and deleted the word “superfood”.
I consider picture number 1 to be unnecessary, because the effects of skupka are better summarized in picture number 2.
Authors response:
We removed the picture number 1. Please note that the figure numbering in the R1 has changed do to the introduction of new graphics.
In picture number 2, quercetin, myricetin and kaempferol should not be listed separately, as they belong to flavonoids. I therefore recommend placing them, for example, in brackets after the term flavonoids. In the same way, the mentioned acids would then deserve the title of phenolic acid.
Authors response:
We applied the suggested change. Please see Figure 6. We also modified the title of phenolic acid.
I also recommend editing the table, deleting the titles that are still repeated in the first column and leaving once antioxidant activity, 1 x antimicrobial, 1 x Antidiabetic, etc. Then leave once methanol extract, ethanol, etc. in the second column. and in the third column then combine the effects for these extracts or form of sumac.
Authors response:
We applied the suggested changes. Please, see the table 2.
/dumbbells should not be written in sentences, but only passwords, as is the case at the end of the table. I would recommend adding a table with the content of specific identified substances, if known, indicating the part of the shell in which the substance was described.
Authors response:
Dumbbells were removed from tables as requested. Regrettably, the part of the shell where the substance was described is not always known and therefore, we cannot add a table.

Reviewer 2 Report
An interesting, a quite well-written review about Sumac is provided. Considering 92 references and various biological, biochemical, and biomedical aspects, the MS is a comprehensive and valuable contribution to attracting further research on this plant species, its potential, and its future use.
However, much information is provided, frequently in wordy paragraphs. Here, more substantive, strategic wording would improve the overall manuscript and, thus, the access and understanding by readers. … For example, lines 60-112 are a wordy text passage that could benefit from focusing only on Sumac.
Some historical aspects would be interesting to mention exploring the bioactive effects of Sumach, i.e., when/for how long the effects of Sumach are known and used, where, and how?
Also, providing insight into botany, one wonders how far the bioactive effects/secondary metabolites matter for the plant (life) because the compounds naturally evolved in the biological context but not for any (human) use. Such a short comment would still perfect the review.
Please see the detailed comments below.
Line 3,
Rhus coriaria
>>> The correct spelling of Rhus coriaria must be checked throughout the complete manuscript. Also, after the first mention of the full name, the Latin species name should be abbreviated as R. coriaria throughout the manuscript – this fact must be checked and needs revision.
Authors and families should be provided for other plants, fungi, and animal species by the first mention.
lines 60-61,
In the globalized modern world, consumers have an ever more deeply rooted belief in the active role that nutrition can have on their state of health.
line 67,
still a "niche",
line 95,
the typical wild Sumac berries
line 109,
literature. This
line 111,
Mediterranean basin, and possibly leading
>>> A review contains a vast pool of data, not only authors' ones.
Figure 1,
>>> Delete the light green box with the text because it is redundant with the Figure legend.
line 121,
Figure 1. Some important pharmaceutical properties of Sumac.
line 123,
R. coriaria L., well-known as Sumac,
line 126,
The genus Rhus comprises more
line 127,
Sumac is a shrub or a small, 1-4-meter high tree distributed in the field/natural habitats/wilderness
OR
Sumac is a shrub or a small tree of 1-4 meters in height and distributed in the field/natural habitats/wilderness
line 130,
dense flower spikes
line 132,
The compound leaves
Figure 2
>>> Scale bars?
line 152,
industry because of its dyeing
line 153,
Dried fruits, grounded into a dark red powder,
line 155,
a flavoring additive
line 169,
The HPLC-MS method identified 191 compounds in R. coriaria, including
Figure 2,
Rhus coriaria
Sumac is a shrub or a small tree of 1-4 meters in height
Red-coloured, densely arranged fruits (drups)
>>> shift the arrow "Phytochemistry "down to the level of "Red-coloured, densely arranged fruits (drups) ", because the information provided right-hand corresponds to the fruits as mentioned in the text above, doesn't it?
in the middle hexagon: some important pharmaceutical properties of Sumac
>>> The images are the same as in Figure 1; thus, redundant. It would be rather professional and informative to add different images/image.
line 173,
Figure 3. Botanical and phytochemical key aspects and pharmaceutical implications of Rhus coriaria (fruits)
line 176,
the highest percentage of Sumac fruits secondary compounds,
lines 189-190,
mg g-1
line 202,
Sumac is a useful plant with
line 208,
effects of Sumac extracts significantly decreased
line 238,
a few pieces of evidence
line 263,
in all human organs
line 269,
studied with 12
line 272,
Bacillus subtilis
line 300,
Recently, several studies
line 310,
conjunctiva, hematosis, hemoptysis, leukorrhea, dermatitis, ocular diseases,
line 361,
Sumac had beneficial
line 362,
where it could prevent
line 363,
tannin had beneficial
line 371,
the consumption
line 405,
macerated ethanol
line 438,
mg ml-1
line 457,
mg kg-1
line 462,
of Sumac Rhus coriaria L.
>>> Table 1 needs a clearer formatting to let readers better follow the row; i.e., rows in each column should clearly correspond to each other, with no mixes.
>>> The Table head should be added on each page, continuing with Table 1.
>>> page 13: The arrows in the table need to be re-formatted, and well fit the rows and columns – currently, they overlay the text and do not correspond to any cited fact.
line 472,
There are a few studies
line 477,
Authors reported genetic
line 489,
program with this species
line 493,
in the Rhus
line 504,
Many fascinating plants
line 506,
Fruits, flowers, leaves, and
>>> How about seeds and roots?
line 513,
(Sumac), its botany
line 522,
towards the domestication
line 524,
agronomic processes
lines 518-525
>>> This is a very, very long sentence, which is hard to follow. It is suggestive of revising, shortening, and clarifying it.
line 526,
Authors' Contribution
Reference list
>>> Latin species names in Italics
Author Response
Reviewer 2:
An interesting, a quite well-written review about Sumac is provided. Considering 92 references and various biological, biochemical, and biomedical aspects, the MS is a comprehensive and valuable contribution to attracting further research on this plant species, its potential, and its future use.
Authors response:
We thank the reviewer for appreciating the manuscript and its style.
However, much information is provided, frequently in wordy paragraphs. Here, more substantive, strategic wording would improve the overall manuscript and, thus, the access and understanding by readers. … For example, lines 60-112 are a wordy text passage that could benefit from focusing only on Sumac.
Authors response:
Following the input of the reviewer, we reshaped out Introduction. Iyìt was reduced from around 880 words to around 480 to better match our content the audience already accustomed to disciplines related to temperate to tropical horticulture.
Some historical aspects would be interesting to mention exploring the bioactive effects of Sumach, i.e., when/for how long the effects of Sumach are known and used, where, and how?
Authors response:
We added the suggested historical information in L
Also, providing insight into botany, one wonders how far the bioactive effects/secondary metabolites matter for the plant (life) because the compounds naturally evolved in the biological context but not for any (human) use. Such a short comment would still perfect the review.
Authors response:
The point of review is attractive but plant-human co-evolution is a subject well beyond the scope of our contribution. In addition, studies on this subject on sumac are to our knowledge, absent. Moreover, this evolutionary topic is probably of wider interest for readers of journals different from Horticulturae. The co-evolution of secondary metabolites involves aspects ranging from plant-plant interaction, allelopathy, plant-insect interactions, parasitism, mutualism, and induction of plant protection by other oorganisms and may deserve a dedicated review if not a book.
Please see the detailed comments below.
Line 3, Rhus coriaria >>> The correct spelling of Rhus coriaria must be checked throughout the complete manuscript. Also, after the first mention of the full name, the Latin species name should be abbreviated as R. coriaria throughout the manuscript – this fact must be checked and needs revision.
Authors response:
We checked the spelling and the use of the abbreviations for Rhus coriaria throughtout the manuscript.
Authors and families should be provided for other plants, fungi, and animal species by the first mention.
Authors response:
We added the authors and families of all the organisms that were cited in the manuscript.
lines 60-61, In the globalized modern world, consumers have an ever more deeply rooted belief in the active role that nutrition can have on their state of health.
Authors response:
Following the request of the reviewer 1 the whole paragraph was removed and the introduction reshaped.
line 67, still a "niche",
Authors response:
During the revision of the manuscript, this sentence was removed.
line 95, the typical wild Sumac berries; line 109, literature. This; line 111,
Mediterranean basin, and possibly leading; >>> A review contains a vast pool of data, not only authors' ones.
Authors response:
This sentences were modified and the text was changed.
Figure 1, >>> Delete the light green box with the text because it is redundant with the Figure legend; line 121: Figure 1. Some important pharmaceutical properties of Sumac.
Authors response:
During the revision of the manuscript, this figure was removed.
line 123, R. coriaria L., well-known as Sumac,
Authors response:
During the revision of the manuscript, this sentence was rephrased.
line 126, The genus Rhus comprises more
Authors response:
Done. (Line 150).
line 127, Sumac is a shrub or a small, 1-4-meter high tree distributed in the field/natural habitats/wilderness OR Sumac is a shrub or a small tree of 1-4 meters in height and distributed in the field/natural habitats/wilderness
Authors response:
The sentence was rephrased (Lines 152-153).
line 130, dense flower spikes
Authors response:
We modified the manuscript accordingly.
line 132, The compound leaves
Authors response:
During the revision of the manuscript, this sentence was removed.
Figure 2>>> Scale bars?
Authors response:
We could not add scale bars in the figure, but we added information on the fruit size directly in the text (Line 154-161).
line 152, industry because of its dyeing
Authors response:
Done (Line 175).
line 153, Dried fruits, grounded into a dark red powder,
Authors response:
The sentence was rephrased (Line 176).
line 155, a flavoring additive
Authors response:
The sentence was rephrased (Line 179).
line 169, The HPLC-MS method identified 191 compounds in R. coriaria, including
Authors response:
During the revision of the manuscript, this sentence was removed.
Figure 2, Rhus coriariaSumac is a shrub or a small tree of 1-4 meters in height Red-coloured, densely arranged fruits (drups) >>> shift the arrow "Phytochemistry "down to the level of "Red-coloured, densely arranged fruits (drups) ", because the information provided right-hand corresponds to the fruits as mentioned in the text above, doesn't it? in the middle hexagon: some important pharmaceutical properties of Sumac>>> The images are the same as in Figure 1; thus, redundant. It would be rather professional and informative to add different images/image.
Authors response:
Following the input, this figure was fully redesigned (new Figure 6).
line 173, Figure 3. Botanical and phytochemical key aspects and pharmaceutical implications of Rhus coriaria (fruits)
Authors response:
The caption of the new Figure 6 was rephrased also taking this comment into account.
line 176, the highest percentage of Sumac fruits secondary compounds,
Authors response:
The sentence was rephrased (Line 200-201).
lines 189-190, mg g-1
Authors response:
Done (Line 213-214).
line 202, Sumac is a useful plant with
Authors response:
The sentence was rephrased (Line 226).
line 208, effects of Sumac extracts significantly decreased
Authors response:
The sentence was rephrased (Line 232-233).
line 238, a few pieces of evidence
Authors response:
We removed this part.
line 263, in all human organs
Authors response:
The sentence is about rat not human (line 270).
line 269, studied with 12
Authors response:
The sentence was rephrased (Line 277-279).
line 272, Bacillus subtilis
Authors response:
We removed this part.
line 300, Recently, several studies
Authors response:
Done. The sentence was reworded (Line 315).
line 310, conjunctiva, hematosis, hemoptysis, leukorrhea, dermatitis, ocular diseases,
Authors response:
We removed this sentence.
line 361, Sumac had beneficial
Authors response:
We removed this sentence.
line 362, where it could prevent
Authors response:
We removed this sentence.
line 363, tannin had beneficial
Authors response:
We removed this sentence.
line 371, the consumption
Authors response:
Done (Line 356).
line 405, macerated ethanol
Authors response:
The sentence was modified (line 403)
line 438, mg ml-1
Authors response:
Done (Line 422).
line 457, mg kg-1
Authors response:
Done (Line 432).
line 462, of Sumac Rhus coriaria L.
Authors response:
We reworded the caption of Table 2.
>>> Table 1 needs a clearer formatting to let readers better follow the row; i.e., rows in each column should clearly correspond to each other, with no mixes.
>>> The Table head should be added on each page, continuing with Table 1. >>> page 13: The arrows in the table need to be re-formatted, and well fit the rows and columns – currently, they overlay the text and do not correspond to any cited fact.
Authors response:
We formatted table 2 so that rows in each column correspond to each other. Mixes are no longer present. We leave the Table head formatting to the editorial office because it will be different in the final version of the manuscript that will be produced (without edits).
line 472, There are a few studies
Authors response:
Done (Line 459).
line 477, Authors reported genetic
Authors response:
We removed this sentence.
line 489, program with this species
Authors response:
We reworded this part (Lines 464-474).
line 493, in the Rhus
Authors response:
Done (Line 475).
line 504, Many fascinating plants
Authors response:
The sentence was rephrased (Lines 497-501).
line 506, Fruits, flowers, leaves, and>>> How about seeds and roots?Authors response:
The sentence was modified to include also the roots (Lines 497-501).
line 513, (Sumac), its botany
Authors response:
The sentence was rephrased (Lines 497-501).
line 522, towards the domestication
Authors response:
The sentence was rephrased (Lines 503-516).
line 524, agronomic processes
Authors response:
The sentence was rephrased (Lines 503-516).
lines 518-525>>> This is a very, very long sentence, which is hard to follow. It is suggestive of revising, shortening, and clarifying it.
Authors response:
The whole part was rewritten (Lines 503-516).
line 526 518, Authors' Contribution
Authors response:
Done.
Reference list>>> Latin species names in Italics
Literature is formatted according to the instructions for authors of the journal.

Round 2
Reviewer 1 Report
I have read the corrected article and I only have comments on Table 2 Antioxidant activity - methanol extract and methanol extract should be combined into one cell in the table, the title methanol extract should not be repeated. What is the difference between dietary sumac and dietary sumac powder, shouldn't these cells also be merged into one? Antidiabetic activity - ethanol extract and ethanol extract 96% have the same literature source, so they are probably the same extracts and could be merged into one cell. Lipid-lowering and hypocholesterolemic activity - Dietary sumac is here three times, it should be merged into one cell. The text in the table is sometimes written in passwords and sometimes in sentences, this needs to be unified. No sentences are necessary for the table.Author Response
Dear Reviewer
In order to match your requests, we have moved Ref 39 above Ref 50. we replaced "Dietary sumac" powder with "Dietary sumac" in Ref 41 and moved Ref 40 and 41 below Ref 50. We also replaced the Ethanol extract with Ethanol 96% (reference 58) and moved them below Ref 14. We replaced "Dietary sumac powder" with "Dietary sumac" in Ref 61 and 67 and moved them below Ref 65. Other minor changes have been highlighted. We have attached the revised paper with the highlighted changes in Table 2. Thanks very much for your consideration,
best regards,
Federico
